# Ethnic Differences in Western and Asian Sacroiliac Joint Anatomy for Surgical Planning of Minimally Invasive Sacroiliac Joint Fusion

**DOI:** 10.3390/diagnostics13050883

**Published:** 2023-02-25

**Authors:** Christopher Wu, Yu-Cheng Liu, Hiroaki Koga, Ching-Yu Lee, Po-Yao Wang, Daniel Cher, W. Carlton Reckling, Tsung-Jen Huang, Meng-Huang Wu

**Affiliations:** 1School of Medicine, College of Medicine, Taipei Medical University, Taipei 110301, Taiwan; 2Department of Orthopedic Surgery, Nanpuh Hospital, Kagoshima 892-0854, Japan; 3Department of Orthopedics, Taipei Medical University Hospital, Taipei 110301, Taiwan; 4Department of Orthopaedics, School of Medicine, College of Medicine, Taipei Medical University, Taipei 110301, Taiwan; 5SI-BONE, Inc., Santa Clara, CA 95050, USA; 6TMU Biodesign Center, Taipei Medical University, Taipei 110301, Taiwan

**Keywords:** sacroiliac joint, anatomy, ethnic differences, transiliac sacroiliac joint fusion

## Abstract

Pain originating in the sacroiliac joint (SIJ) is a contributor to chronic lower back pain. Studies on minimally invasive SIJ fusion for chronic pain have been performed in Western populations. Given the shorter stature of Asian populations compared with Western populations, questions can be raised regarding the suitability of the procedure in Asian patients. This study investigated the differences in 12 measurements of sacral and SIJ anatomy between two ethnic populations by analyzing computed tomography scans of 86 patients with SIJ pain. Univariate linear regression was performed to evaluate the correlations of body height with sacral and SIJ measurements. Multivariate regression analysis was used to evaluate systematic differences across populations. Most sacral and SIJ measurements were moderately correlated with body height. The anterior–posterior thickness of the sacral ala at the level of the S1 body was significantly smaller in the Asian patients compared with the Western patients. Most measurements were above standard surgical thresholds for safe transiliac placement of devices (1026 of 1032, 99.4%); all the measurements below these surgical thresholds were found in the anterior–posterior distance of the sacral ala at the S2 foramen level. Overall, safe placement of implants was allowed in 84 of 86 (97.7%) patients. Sacral and SIJ anatomy relevant to transiliac device placement is variable and correlates moderately with body height, and the cross-ethnic variations are not significant. Our findings raise a few concerns regarding sacral and SIJ anatomy variation that would prevent safe placement of fusion implants in Asian patients. However, considering the observed S2-related anatomic variation that could affect placement strategy, sacral and SIJ anatomy should still be preoperatively evaluated.

## 1. Introduction

Pain originating in the sacroiliac joint (SIJ) is increasingly recognized as an important contributor to chronic low back pain [1,2]. Nonsurgical treatments for SIJ pain are common, yet their long-term efficacy is unknown. Furthermore, nonsurgical treatments are inadequate for some patients with SIJ pain [3]. By contrast, the placement of triangular titanium implants through a lateral transiliac approach has been shown to be a safe and effective treatment of SIJ dysfunction resulting from degeneration or disruption of the SIJ in two randomized trials [4,5], a large multicenter prospective trial [6], and several case series [7,8] and comparative cohorts [9,10,11].

Minimally invasive SIJ fusion with a lateral transiliac approach is an attractive treatment option for patients with SIJ pain. The goal of implant placement is to position devices across the articular portion of the joint and to fully seat these devices in the sacrum, without malposition into the sacral foramen or spinal canal and without violation of the cephalad, ventral, or caudal sacral cortex. However, anatomic variations of the SIJ have been reported, including differences by sex [12], handedness [13], and the degree of degeneration [14]. Studies have also reported sacral body variations relevant to the placement of screws for pelvic fixation after trauma [15,16], including in an Asian population [17,18]. Sacral dysmorphism is also relevant to the placement of surgical devices [19,20,21].

Currently, all published studies on minimally invasive SIJ fusion for treating chronic pain have been performed in Western populations. Given the known shorter stature of Asian people compared with Westerners, concerns can be raised regarding the applicability of the current evidence base to Asian populations. In the present study, we evaluate sacral and SIJ anatomic variations as they relate to minimally invasive lateral transiliac device placement during minimally invasive SIJ fusion.

## 2. Materials and Methods

### 2.1. Patient Population and Assessment

Pelvic computed tomography (CT) scans of symptomatic patients with SIJ pain were obtained from two sources: (1) the Department of Orthopedic Surgery, Kikuno Hospital, Kagoshima, Japan, and (2) two multicenter US clinical trials on minimally invasive SIJ fusion (INSITE, a prospective randomized controlled trial, and SIFI, a prospective single-arm trial). Diagnoses in all cases were made after history taking, physical examination, tests that stressed the SIJ [22], and diagnostic SIJ blocks with local anesthetics that produced marked acute relief of typical pain. This study was approved by the Institutional Review Board of Taipei Medical University Hospital (TMU-JIRB No. N201706063). The need for informed consent was waived by the Research Ethics Committee because the present study was based on retrospective image analysis.

Each CT scan was uploaded to the Materialise Mimics software (version 20.0, Materialise NV, Leuven, Belgium; run on a Windows 10 PC), segmented, and masked to remove the ilium and obtain a stereolithographic (STL) model of the sacrum, focusing on the SIJ and the lateral sacral surface. Using the STL model, we measured the inferior and superior articular limb distances (Table 1 and Figure 1) and the lateral SIJ surface area in millimeters squared (measured using Materialise 3-Matic Research, version 12.0). Multiplanar reconstruction parallel to the long axis of each sacrum was performed to make the remaining two-dimensional measurements (Table 1). In total, 12 measurements were used for evaluation of sacral and SIJ anatomy. For all patients, scans of both sides were segmented and measured.

### 2.2. Data Set Preparation and Analysis

Two data sets were prepared and analyzed as follows.

#### 2.2.1. Reproducibility

Scans from 10 randomly selected patients were analyzed twice at least 1 week apart by the same reviewer. Measurement reproducibility was assessed using Bland–Altman plots. The observed accuracy was 4–5 mm for the superior and inferior articular limb, 200 mm^2^ for the surface area, and 2–4 mm for measurements made from the multiplanar reconstruction views. These corresponded to coefficients of variation of approximately 10–20%, which were deemed sufficient for this analysis.

#### 2.2.2. Symmetry

To assess SIJ symmetry, a paired right–left data set (*n* = 86 patients, 172 sides) was prepared to enable graphical and statistical analyses of symmetry within individuals across SIJ sides.

#### 2.2.3. Primary Analysis Data Set

The primary focus of our analysis was the correlations of total body height with sacral and SIJ measurements as well as the cross-ethnic differences in these measurements. To preserve statistical independence, the main analysis data set consisted of one side only per patient (Table 2). For Asian patients with unilateral symptoms, the symptomatic side was selected unless it contained an implant, in which case the asymptomatic side was selected. For Asian patients with bilateral symptoms, the side was selected at random. For Western patients, the implant side was selected.

#### 2.2.4. Distance Sufficiency

Using the main primary analysis data set, the 12 measurements for each patient were compared with threshold measurements determined on the basis of implant size and surgical considerations for the successful lateral transarticular placement of triangular titanium implants (iFuse Implant System, SI-BONE, Inc., Santa Clara, CA, USA), which are the most well-studied device used for SIJ fusion (Table 3). For cases in which measured sacral distances were below the threshold, the effect on device placement strategy was further assessed.

#### 2.2.5. Body Height Distribution

The population distributions of body height were obtained from national surveys. Body height data were available for all analyzed patients.

### 2.3. Statistical Analysis

Right–left SIJ symmetry was evaluated using Pearson’s correlation coefficient. The correlations between individual variables (of greatest interest was total body height) and sacral/SIJ measurements were evaluated using Pearson’s correlation coefficient and univariate linear regression applied to the primary analysis data set. Multivariate linear regression was used to estimate the effect of sex and ethnicity on sacral and SIJ measurements. All statistical analyses were performed using R [23].

## 3. Results

Relatively high degrees of right–left symmetry were discovered for all sacral and SIJ measurements relevant to the placement of triangular titanium implants through the transiliac approach; we found a small number of outliers indicating expected anatomic right–left variation within individuals (data not shown). The Pearson coefficients of correlation between right and left sacral and SIJ measurements ranged from 0.49 to 0.86 (S1 body midline to lateral sacral cortex and SIJ surface area, respectively; data not shown). Dysmorphic sacra (sacralization of L5 and other anatomic variants) were commonly seen.

The correlations of body height with sacral and SIJ anatomic measurements were moderate for most but not all measurements, and the correlations were more likely to be significant in men than in women (Table 4). In women, the linear relationships between total body height and sacral and SIJ measurements were modest, with the differences being approximately 1 mm with every centimeter increase in body height in 7 of the 12 measurements (Table 5). The difference in SIJ measurements between an Asian woman in the 1st percentile (1.43 m) versus one in the 99th percentile (1.66 m) of height ranged from 0 to 8 mm (Table 5). In men, all sacral and SIJ measurements but one correlated with body height, resulting in a larger range (3 to 12 mm) across the height percentile extremes (1st to 99th percentile; Table 5).

Body height, sex, and ethnicity were then included in multivariate general linear models to investigate systematic differences in sacral and SIJ measurements as a function of these variables. Controlled for body height, men had slightly larger measurements for the inferior articular limb and SIJ height as well as smaller lateral measurements (midline to the sacral cortex and lateral border of the foramen to the sacral cortex; Table 6). These findings correspond to the well-described sacral sexual dimorphism—women having wider sacra medially to laterally.

Compared with Western patients and after controlling for body height, one measurement (AP thickness of the sacral ala at the level of the S1 body) was slightly smaller in Asian patients (average = 3.0 mm) and two measurements (S1 body midline to the lateral sacral cortex and S1 lateral foramen to the lateral sacral cortex) were slightly larger (by 2.2 and 2.3 mm, respectively; Table 7). In further analysis stratified by sex, the AP thickness of the sacral ala at the level of the S1 body was significantly smaller in Asian women but not Asian men (by 3.5 mm, *p* = 0.0110, and by 1.2 mm, *p* = 0.6479, respectively; data not shown).

As illustrated in Figure 2, only a small number of measurements (6 of 1032, 0.6%; 6 of 86 patients, 6.98%, three from each geographic region) were below the surgical thresholds defined in Table 3; all these measurements were for the AP thickness of the sacral ala at the level of the S2 foramen and occurred in three women and three men. Of these six cases, further assessment by the manufacturer of the iFuse Implant System revealed that sacral and SIJ anatomy could nonetheless accommodate three implants in four cases (4.7% of all cases); however, only two implants were possible in the remaining two cases (2.3% of all cases; see Discussion section for our explanation). Overall, safe placement of three triangular titanium implants was allowed in 84 of 86 (97.7%) of our included patients. Regression analysis revealed no correlation of body height or ethnicity with whether a measurement was below the surgical threshold.

## 4. Discussion

The present study obtained data with which to examine the following questions relative to sacral and SIJ anatomy: (1) To what extent is total body height correlated with sacral and SIJ measurements relevant to transiliac placement of permanent implants across the SIJ? (2) Taking the known overall shorter stature of Asian populations into account, are there systematic differences in SIJ measurements between Western and Asian populations that would make implant placement challenging in Asian patients, lead to fewer devices being used, or make the development of smaller devices worthwhile? The second question may be of most relevance to Asian women, who are shorter on average compared with Western men; in addition, women are more susceptible to SIJ dysfunction than men [22,24]. In our analyses, the correlations of body height with sacral and SIJ measurements were modest; only minor differences in sacral and SIJ anatomic measurements were discovered across populations. Overall, neither body height nor ethnicity was correlated with below-threshold sacral and SIJ measurements.

Our study revealed that correlations of body height with sacral and SIJ measurements were moderate in women and slightly stronger, on average, in men (Table 4 and Table 5). This finding may suggest more cross-individual SIJ anatomic variations in women compared with in men. After controlling for body height, we discovered a few systematic differences in SIJ anatomic measurements between Asian and Western symptomatic patients (Table 7); only one measurement was slightly smaller: the AP thickness of the sacral ala at the level of the S1 body (3 mm smaller), which was a significant difference in Asian women only. Overall, the cross-population differences were minor. Clinical studies reporting minimally invasive SIJ fusion by using triangular titanium implants have included almost entirely Western patients with a broad range of body heights; subgroup analysis of pooled data from more than 320 participants enrolled in prospective trials revealed no differences in improvement of SIJ pain score or disability score (as assessed using the Oswestry Disability Index) in shorter versus taller patients [25]. Taken together, these findings allow us to conclude that although Asian populations are shorter on average than Western populations, the potential differences in sacral and SIJ anatomy between Asian and Western patients are modest and unlikely to require alterations of device design or placement strategies. Our finding is supported by cross-ethnic studies on other sacral orthopedic applications and suggesting modest differences between Asian and Western populations [26].

We measured the sacral ala in the medial–lateral and AP directions at four cranial–caudal levels: at the level of the S1 body (above the S1 foramen), the S1 foramen, the S2 body (between the first and second neuroforamen), and the S2 foramen. As illustrated in Figure 2, the AP thicknesses at the level of the S2 foramen were below defined surgical thresholds for safe placement of triangular titanium implants in only six cases (6/86, 6.98%). Similar to how variations in sacral anatomy and sacral segmentation can affect ilio–sacral screw placement [7,16,27], we observed sacral variation that can potentially influence placement strategies for triangular titanium implants (and perhaps other implants placed through the minimally invasive lateral transiliac route). After assessment by the manufacturer of the iFuse Implant System, placement of three triangular titanium implants was allowed in four of the six aforementioned cases; overall, nearly all of our included patients (84 of 86, 97.7%) could safely receive the implants.

Typically, three triangular titanium implants are placed, the first deeply (“long”) at the level of the S1 body, the second more shallowly (“short”) and laterally at the level of the S1 foramen, and the third deeply (“long”) at the level of the S2 body (i.e., a long–short–long approach). Occasionally, anatomic variations (e.g., sacralization of the L5 vertebra) prevent implant placement at the level of the most cephalad sacral body; to place three implants in this anatomic configuration, the first implant is placed “short” at the level of the S1 foramen, the second implant is placed “long” at the level of the S2 body, and the third implant is placed “short” into the sacrum at the level of the S2 foramen (i.e., a short–long–short approach). In the six aforementioned cases in which the AP thickness at the level of the S2 foramen was below the threshold, sacral and SIJ anatomy in four of the six cases enabled placement of implants in the long–short–long configuration; in these cases, the below-threshold S2 medial–lateral or AP thicknesses were not relevant (as no implant would be placed at this level). However, for the remaining two cases, a dysmorphic sacrum combined with a relatively small AP sacral ala distance at the S2 body level would have required a short–long (i.e., two-implant) approach. The manufacturer recommended placement of three implants for optimal stabilization and increased the implant surface area for osseous integration; whether clinical responses are less favorable with placement of only two implants is unknown. In a pooled analysis of prospective trials of the same device, only a few cases (11/326, 3.4%) involved the use of only two implants, making it challenging to draw any conclusions regarding the efficacy of two versus three implants [25]. Overall, we discovered no meaningful differences in body height that distinguished these two patients with below-threshold distances at the S2 level precluding implant placement at this level.

Sacral and SIJ anatomic variations that may affect the placement of surgical devices have been reported by numerous scholars [15,16,17,18,19,20,21]. In our study, we observed minor cross-ethnic differences in sacral and SIJ anatomic measurements (smaller AP thickness at the level of the S1 body in Asian patients). We also found both within-individual and cross-individual differences in the dimensions of the sacral ala at the level of the S1 and S2 sacral bodies. Moreover, we commonly observed dysmorphic sacral features. However, our regression analysis revealed no correlations between ethnicity and below-threshold sacral and SIJ measurements; overall, as shown in Figure 2, despite sacral and SIJ anatomic variations being discovered, they generally had minimal effect on the placement strategy for fusion implants placed in the sacral ala through the lateral transiliac route.

Our analysis focused on sacral and SIJ measurements related to but slightly different from those used in studies investigating screw placement across the SIJ into the S1 and S2 sacral bodies during surgery for traumatic pelvic fracture. Nonetheless, many of our measurements were in ranges similar to those reported previously from both Japan [28] and the United States [29]. One strength of our study is that our findings may be generalizable to other devices placed across the SIJ when a lateral transiliac approach is used. Other strengths include its analyses of symptomatic patients with SIJ pain from two ethnicities (Westerners and Asians); ours is the first study to compare these two populations. Furthermore, we used software that has been employed in previously published analyses of SIJs in Asian populations [10].

Some limitations of our study need to be addressed, however. First, as mentioned, the sacral and SIJ measurements used in our study were mainly employed for evaluation of the lateral transiliac placement of fusion implants; therefore, our analysis is not applicable to devices placed through a different (e.g., posterior or posterolateral) surgical route. Second, our analysis was based on anatomy determined through CT, which might be different from that discovered through surgical radiographic imaging. Third, we did not group measurements by the available sacral configuration systems (e.g., Mahato et al. [30]), despite the fact that sacral configuration can, in some cases, affect implant placement. Fourth, we only included Eastern Asian patients in our study; these patients may not be representative of other Asian populations. Our findings regarding the minor cross-ethnic differences in sacral and SIJ anatomic measurements should be extrapolated with caution.

## 5. Conclusions

Sacral and SIJ anatomy relevant to transiliac device placement is variable and correlates modestly with body height. When body height is controlled, cross-ethnic variations in sacral and SIJ anatomy are minor. Our findings raise a few concerns regarding sacral and SIJ anatomic variation that would prevent safe placement of fusion implants in Asian patients. However, considering our finding regarding the S2-level anatomic variations that can influence device placement, sacral and SIJ anatomy should still be evaluated preoperatively, particularly through measurements at the S2 level, to determine the safe placement of transiliac implants.

## Figures and Tables

**Figure 1 diagnostics-13-00883-f001:**
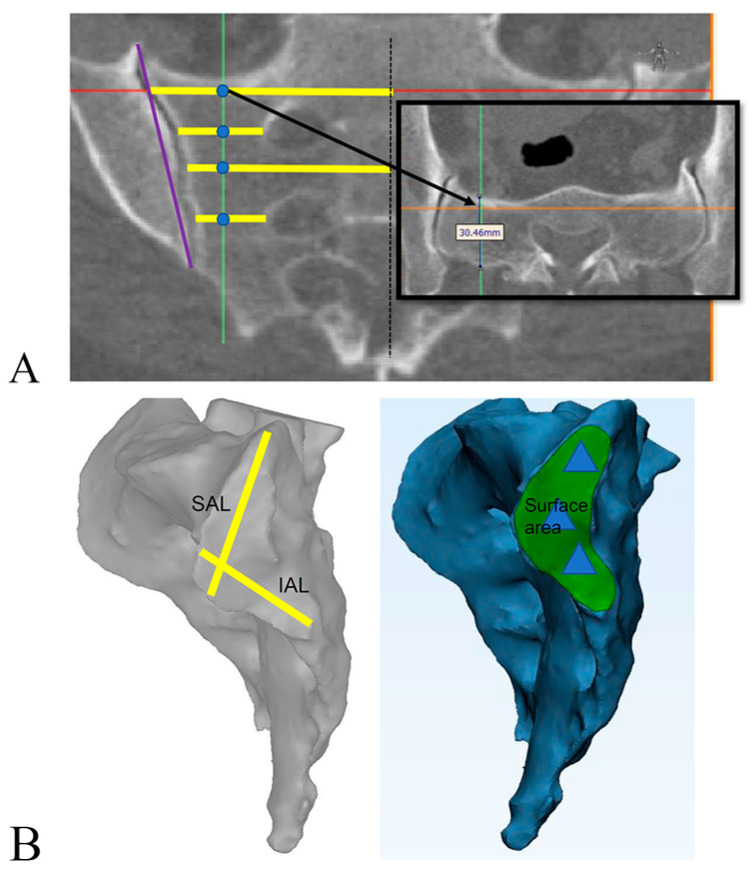
Pictorial representation of sacral and sacroiliac joint measurements. (**A**) Measurements made in medial–lateral (yellow line), cranial–caudal (purple line), and anterior–posterior directions (black box; inset, at level of S1 body). Blue dots show medial–lateral and anterior–posterior ala measurements at the levels of the S1 body, S1 foramen, S2 body, and S2 foramen. Red line: upper margin of sacrum; green line: middle line of sacral ala; orange line: anterior margin of sacrum. (**B**) Definition of SAL (yellow line, upper), IAL (yellow line, lower), and surface area (green area) on segmented scans. Blue triangles represent typical positions for triangular titanium implants. All measurements are described in detail in Table 1. Abbreviations: IAL, inferior articular limb; SAL, superior articular limb.

**Figure 2 diagnostics-13-00883-f002:**
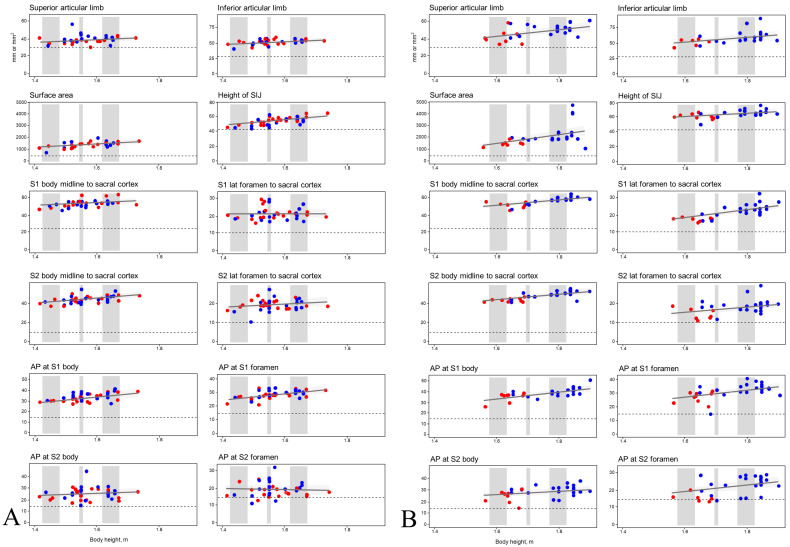
Correlations between body height and sacral and SIJ measurements in women (**A**) and men (**B**). Blue: Western patients; red: Asian patients. Gray bars represent 1st–10th percentile, 45th–55th percentile, and 90th–99th percentile of total body height. Abbreviations: AP, anterior–posterior; lat, lateral; SIJ, sacroiliac joint.

**Table 1 diagnostics-13-00883-t001:** Sacral and SIJ measurements assessed in this study.

**Measured using a segmented STL model** Length of SAL measured from the apex of the SIJ to the anterior cortex, bisecting the articular limb. Length of IAL measured from the base of the SIJ to the superior surface, bisecting the bottom half of the sacrum. SIJ surface area.
**Measured through multiplanar reconstructions parallel to the S1 body** 4. Distance from the lateral border of the S1 foramen to the lateral cortex of the sacrum. 5. Distance from the lateral border of the S2 foramen to the lateral cortex of the sacrum. 6. Distance from the midline of the S1 body to the lateral cortex of the sacrum. 7. Distance from the midline of the S2 body to the lateral cortex of the sacrum. 8. Height of the SIJ.
**Sacral ala at the midline between the foramen and the lateral border of the SIJ** 9. Anterior–posterior thickness at the level of the S1 body. 10. Anterior–posterior thickness at the level of the S1 foramen. 11. Anterior–posterior thickness at the level of the S2 body. 12. Anterior–posterior thickness at the level of the S2 foramen.

Abbreviations: IAL, inferior articular limb; SAL, superior articular limb; SIJ, sacroiliac joint.

**Table 2 diagnostics-13-00883-t002:** Number and baseline characteristics of analyzed patients.

	Female		Male		Total		Baseline Characteristics		
Ethnicity	Patients	Sides	Patients	Sides	Patients	Sides	Age (Years; Mean ± SD)	Height (cm; Mean ± SD)	Weight (kg; Mean ± SD)
**Western**	25	50	22	44	47	94	64.8 ± 11.2	178.2 ± 13.6	77.5 ± 23.9
**Asian**	23	46	16	32	39	78	69.2 ± 13.8	170.8 ± 7.5	62.3 ± 12.6

**Table 3 diagnostics-13-00883-t003:** Derivation of surgical thresholds used in this study.

Measurement	Threshold	Rationale
Superior articular limb	28.4 mm	Distance across inscribed diameters of two implants. Two implants are typically placed in the superior limb of the SIJ.
Inferior articular limb	28.4 mm	Distance across inscribed diameters of two implants. Two implants are typically placed in the inferior limb of the SIJ. Note that second implant is in both limbs.
Surface area	3 × π*r^2^ = 475 mm^2^	Cross-sectional area of circles formed by three implants.
Lateral border of the S1 foramen to the lateral cortex	10 mm	10 mm engagement depth represents minimal implant engagement likely resulting in immediate stabilization.
Lateral border of the S2 foramen to the lateral cortex		
S1 body midline to the lateral cortex of the sacrum		
S2 body midline to the lateral cortex of the sacrum		
SIJ height	42.6 mm	Total length of three diameters of implants.
AP thickness of the sacral ala	14.2 mm	The AP thickness should be larger than the inscribed diameter of the implant; otherwise, the implant protrudes from the sacrum.

Abbreviations: AP, anterior–posterior; SIJ, sacroiliac joint.

**Table 4 diagnostics-13-00883-t004:** Correlation between body height and sacral and SIJ measurements.

Measurement *	Sex			
	Women		Men	
	R **	*p* Value	R **	*p* Value
Superior articular limb	0.17	0.1691	0.52	0.0003
Inferior articular limb	0.29	0.0155	0.54	0.0002
Surface area	0.26	0.0315	0.49	0.0008
Height of the SIJ	0.42	0.0003	0.38	0.0093
S1 body midline to the sacral cortex	0.38	0.0013	0.60	<0.0001
S1 lateral foramen to the sacral cortex	0.02	0.8400	0.59	<0.0001
S2 body midline to the sacral cortex	0.42	0.0003	0.50	0.0003
S2 lateral foramen to the sacral cortex	0.11	0.3746	0.45	0.0015
AP at the S1 body	0.31	0.0097	0.67	<0.0001
AP at the S1 foramen	0.28	0.0208	0.44	0.0019
AP at the S2 body	0.05	0.6583	0.23	0.1186
AP at the S2 foramen	−0.08	0.5298	0.36	0.0134

Abbreviations: AP, anterior–posterior thickness; SIJ, sacroiliac joint. * The unit for all measurements is mm except for surface area, for which it is mm^2^. ** r: Pearson’s correlation coefficient.

**Table 5 diagnostics-13-00883-t005:** Univariate linear relationship between total body height and sacral and SIJ measurements.

	Women					Men				
Measurement *	Regression Coefficient **	*p* Value	Predicted 1 ***	Predicted 99	Range ****	Regression Coefficient **	*p* Value	Predicted 1 ***	Predicted 99	Range
SAL	0.14	0.1691	37.2	40.4	3.2	0.40	0.0003	41.0	51.0	10.0
IAL	0.23	0.0155	47.1	52.3	5.2	0.48	0.0002	53.2	65.2	12.1
Surface area	9.21	0.0315	1211.3	1421.3	210.0	36.10	0.0008	1385.1	2288.8	903.7
Height of the SIJ	0.36	0.0003	50.0	58.2	8.1	0.20	0.0093	60.2	65.1	4.9
S1 body midline to the sacral cortex	0.24	0.0013	52.5	58.0	5.5	0.29	<0.0001	52.1	59.5	7.4
S1 lateral foramen to the sacral cortex	0.02	0.8400	21.9	22.2	0.4	0.21	<0.0001	17.9	23.3	5.4
S2 body midline to the sacral cortex	0.27	0.0003	41.2	47.4	6.2	0.22	0.0003	41.4	46.9	5.5
S2 lateral foramen to the sacral cortex	0.05	0.3745	16.6	17.7	1.1	0.14	0.0015	13.9	17.5	3.6
AP at the S1 body	0.22	0.0097	28.8	33.8	5.1	0.35	<0.0001	31.3	40.2	8.9
AP at the S1 foramen	0.15	0.0208	24.8	28.3	3.5	0.20	0.0019	26.7	31.8	5.1
AP at the S2 body	0.05	0.6583	23.9	25.0	1.0	0.12	0.1186	25.1	28.1	3.0
AP at the S2 foramen	−0.05	0.5298	20.1	18.9	−1.2	0.16	0.0134	17.6	21.5	3.9

* The unit for all measurements is mm except for surface area, for which it is mm^2^. ** Difference in measurement in mm (or mm^2^) for every cm increase in total body height. *** Model estimate for women or men in the 1st and 99th percentile of body height. **** Model estimate for difference in measurement between 1st and 99th percentile of body height. Abbreviations: AP, anterior–posterior thickness; IAL, inferior articular limb; SAL, superior articular limb; SIJ, sacroiliac joint.

**Table 6 diagnostics-13-00883-t006:** Effect of male sex on sacral and SIJ measurements after controlling for body height.

	Comparison	
	Men vs. Women	
Measurement *	Regression Coefficient **	*p* Value
Superior articular limb	2.8	0.0743
Inferior articular limb	3.8	0.0224
Surface area	142.2	0.2438
Height of the SIJ	4.8	0.0004
S1 body midline to the sacral cortex	−3.5	0.0008
S1 lateral foramen to the sacral cortex	−3.6	0.0004
S2 body midline to the sacral cortex	−3.7	0.0004
S2 lateral foramen to the sacral cortex	−3.0	0.0001
AP at the S1 body	−0.1	0.9420
AP at the S1 foramen	0.0	0.9895
AP at the S2 body	0.8	0.5608
AP at the S2 foramen	−1.1	0.3425

* The unit for all measurements is mm except for surface area, for which it is mm^2^. ** Difference in measurement in mm (or mm^2^) for men compared with women. Abbreviations: AP, anterior–posterior thickness; SIJ, sacroiliac joint.

**Table 7 diagnostics-13-00883-t007:** Effect of ethnicity on sacral and SIJ measurements after controlling for body height.

	Comparison	
	Asian vs. Western Patients	
Measurement *	Regression Coefficient **	*p* Value
Superior articular limb	−1.9	0.2425
Inferior articular limb	−0.2	0.9207
Surface area	5.0	0.9714
Height of the SIJ	1.5	0.2596
S1 body midline to the sacral cortex	2.2	0.0114
S1 lateral foramen to the sacral cortex	2.3	0.0245
S2 body midline to the sacral cortex	0.4	0.7076
S2 lateral foramen to the sacral cortex	−0.3	0.6675
AP at the S1 body	−3.0	0.0109
AP at the S1 foramen	−0.4	0.7171
AP at the S2 body	−1.5	0.3158
AP at the S2 foramen	0.2	0.8912

* The unit for all measurements is mm except for surface area, for which it is mm^2^. ** Difference in measurement in mm (or mm^2^) for Asian patients compared with Western patients. Abbreviations: AP, anterior–posterior; SIJ, sacroiliac joint.

## Data Availability

No new data were created or analyzed in this study. Data sharing is not applicable to this article.

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
