# Peer review of "Ethnic Differences in Western and Asian Sacroiliac Joint Anatomy for Surgical Planning of Minimally Invasive Sacroiliac Joint Fusion"

_diagnostics, 2023, doi:10.3390/diagnostics13050883_

Round 1

Reviewer 1 Report

The manuscript “Ethnic Differences in Asian and Caucasian Sacroiliac Joint Anatomy for Surgical Planning of Minimally Invasive Sacroiliac Joint Fusion” by  Christopher Wu et al. aimed to evaluate sacral/SIJ anatomy variation as it relates to minimally invasive lateral transiliac device placement during minimally invasive SIJ fusion.

Below are my comments and remarks regarding the manuscript:

1. Were the results consistent with the normal distribution, if not, it is better to use the Spearman's coefficient. ,

2. Table 2 - Please provide the range or SD for the weight height age value

3. Table 2. Pateints -> Patients

4. Table 5, please describe the Coefficient in the methodology in more detail.

5. Table 5 Why In Surface area Coefficient was greater than one? Why are RR and 95CI not given?

Author Response

Response to Reviewer 1 comments:

Point 1: Were the results consistent with the normal distribution, if not, it is better to use the Spearman's coefficient. 

Response 1: Our raw data regarding the body height, weight, and age were in normal distribution, thus we evaluated the correlation between these parameters and sacral/SIJ measurements using Pearson’s correlation coefficient.

Point 2: Table 2 - Please provide the range or SD for the weight height age value

Response 2: We have added the above information in our revised manuscript. Thank you for your reminder!!!

Point 3: Table 2. Pateints -> Patients

Response 3: Thank you for your reminder. We have corrected this in our revised manuscript.

Point 4: Table 5, please describe the Coefficient in the methodology in more detail.

Response 4: Same response as Point 5; we have revised our manuscript.

Point 5: Table 5 Why In Surface area Coefficient was greater than one? Why are RR and 95CI not given?

Response 5: We are sorry for the misleading word “coefficient” in our tables. The “coefficient” in table 5 means “regression coefficient”, which was obtained through univariate linear regression analysis. For surface area, the provided “coefficients” mean an increase of 9.21 and 36.1 mm2 with every 1 cm increase in body height in women and men, respectively. We have corrected the word “coefficient” to “regression coefficient” in our manuscript.

Reviewer 2 Report

Many thanks for your email and please accept my apologies for the long delay in processing your manuscript.

The paper is well organized, but there are some issues to clarified. 

-       Table 2: I think that the title “mean” was missing in the 4th column. Moreover, in table 2 must be added the standard deviation in the age, weight and height column.

-       Results and discussion are not well organized, and the message is not clear.

-       One group is Caucasian and the second is Asian people. Do you think that this classification may be valid in 2022? Dividing the population for some parameters (sex, high, weight, ecc) may be more accurate than for “origin”.

I’m afraid but the paper must underwent major revision. 

Author Response

Response to Reviewer 2 comments:

Point 1: Table 2: I think that the title “mean” was missing in the 4th column. Moreover, in table 2 must be added the standard deviation in the age, weight and height column.

Response 1: Thank you for your reminder. We have corrected and added the required information in our revised manuscript.

Point 2: Results and discussion are not well organized, and the message is not clear.

Response 2: We have reorganized our manuscript. Thank you for your advice!!!

Point 3: One group is Caucasian and the second is Asian people. Do you think that this classification may be valid in 2022? Dividing the population for some parameters (sex, high, weight, ecc) may be more accurate than for “origin”

Response 3: Thank you for your comment. There are anatomic variations which could affect the placement of surgical devices at SIJ in Asian population [1, 2]. In our opinion, considering that minimally invasive SIJ fushion has been well studied in Caucasians but not in Asians, potential anatomic variations (which can be cross-ethnic or cross-individual) may still affect the safety and placement strategy regarding the procedure when performed in Asians.

  1. Kwan, M.K., et al., A radiological evaluation of the morphometry and safety of S1, S2 and S2-ilium screws in the Asian population using three dimensional computed tomography scan: an analysis of 180 pelvis. Surg Radiol Anat, 2012. 34(3): p. 217-27.
  2. Tian, X., et al., Morphometry of iliac anchorage for transiliac screws: a cadaver and CT study of the Eastern population. Surg Radiol Anat, 2010. 32(5): p. 455-62.

Reviewer 3 Report

The  authors report on the  anatomy  and  fusion  of  SIJ   for low  back pain  in an Asian population compared to Caucasians in  86 symptomatic patients (47 Caucasian, 39 Asian). On the  basis  of plain roentgenograms  and  CT scans of 12 measurements of  SIJ anatomy was performed . Differences across the  two ethnic  populations were recorded as regard  for total body height and gender. Most sacral/SIJ measurements were moderately correlated with total body height. The anterior-posterior (AP) thickness of sacral ala at the level of S1 body was smaller in Asian women compared to Caucasian women. Most measurements were above standard surgical thresholds for safe transiliac placement of devices; all measurements below surgical thresholds were found in AP distance of the sacral ala at S2 foramen level. Sacral/SIJ anatomy relevant to transiliac device placement is variable and correlates modestly with body height. The  authors concluded that because of  the  shorter stature of Asian populations,  S2 anatomy must be evaluated preoperatively to determine the safe placement of transiliac implants.

Interesting  anatomical  study because  of  increasing  SIJ  fusion rates  all over  the  world.

The use of MIS  lateral transiliac device placement for minimally invasive SIJ fusion is an attractive treatment option for SIJ pain.

Differences  of  sacrum anatomy between  genders   have  already  published.

Publications reports  in Asian population sacral body variations relevant to placement of screws for pelvic fixation after trauma have  been done.

The  vast  majority of  the  already published papers regarding   SIJ fusion for chronic pain has been performed  in Caucasian populations. The  authors  have  logically asked if  the  dimensions of  iliosacral joint  and  sacrum  are  different  in  Asian populations given the known of shorter stature of Asian populations compared to Caucasians.

The  authors  have  correctly presented  the  data and  methods  in  the  Methods  section. Reproducibility was  correctly made as it  is  an important  measure  for  repeatability of  the  methods.

The  statistical methods  are  correctly performed.

Although it  is  obvious  and  logic the  Asia  sacral bone  dimensions should  be  smaller  that  the  Caucasians  dimensions, this  excellent  study  provided  useful  information  for  surgeons operating  Asian patients. Special sizes  of  the  fusion implants  should  be  available.

Author Response

Response to Reviewer 3 comments:

Thanks for your comments. We have revised our manuscript and sent it for English editing. Thank you for your time and advice!!!

Reviewer 4 Report

Assessment of the this paper is below (coments are highlighted with yellow) 

Surgery related to diverse forms of chronic bones, joints, tendons and ligaments pathologies should be orchestrated in accordance with the clinical phenotype of each patient.  I read this paper with care and I concluded that the authors' main interest is directed towards a segmental treatment of their patients.  It says that the authors are more technically oriented than towards genuine clinical practice. In orthopedic surgery research, results are gradually accumulated and supposedly invested into profitable clinical care, and if there is no solid connection between effective clinical practice and orthopedic research, the outcome is mostly awkward and meaningless.  Perhaps more surprisingly, that despite the huge amount of published papers in orthopedic research, this does not necessarily lead to improved patient care.  Simply because, the vast majority of these papers are based on segmental/technical repair. Thereby, lose the capability to fill the gap between the true clinical practice and treatment. Authors need to immerse themselves more into comprehensive clinical documentation and remember that human body cannot partitioned.

Pelvic CT scans for symptomatic patients diagnosed with SIJ pain were obtained

Comment: Please show AP pelvis radiographs of at least six patients (Asian and Caucasian).

Also 3D reconstruction Pelvis CT scan.

Line 103: Pictorial representation of sacral/sacroiliac joint measures

Comment: This can never replace the etiology understanding of SIJ disorders.  

Line 195-196: This study provided data to examine the following hypotheses relative to sacral/SIJ  anatomy: 1) to what extent are SIJ measurements relevant to transiliac placement of permanent implants across the SIJ related to total body height.

Comment: Totally disagree, any sacral/SIJ anatomy disruption is almost always linked to other anatomical structures. It´s of extreme rarity to encounter SIJ anatomical disruption/malformation in isolation with other skeletal problems.   

Line 281: Sacral/SIJ anatomy relevant to transiliac device placement is variable and correlates modestly with body height.

Comment: Disagree, patients with tall stature regardless the race are manifesting SIJ problems as much as the short stature subjects. At the end, it’s almost always strongly correlated with the clinical phenotype of each patient. 

Author Response

Please see the attachment "Response to Reviewer 4 comments"

Round 2

Reviewer 1 Report

I have no more comments

Author Response

Thank you for your time in reviewing our manuscript and your comments!!!

Reviewer 4 Report

Revising the AP pelvis radiographs

Patient 1 and 2 is the same patient (delete 1) 

Patient 2 AP plevis radiograph shows mild coxa vara in connection with small capital femoral epiphyses (dysplastic) and abbreviated femoral necks (give the age and the height of the patient)

Patient 3 and 5 same patient : Age? seems osteopenic 

Patient 4 , give age? very likely syndromic-delete 

Pateint 6 asymmetry ? Lower limb length asymmetry? 

Author Response

Point:

Patient 1 and 2 is the same patient (delete 1) 

Patient 1 and 2 are different persons. We have deleted the image of patient 1.

Patient 2 AP plevis radiograph shows mild coxa vara in connection with small capital femoral epiphyses (dysplastic) and abbreviated femoral necks (give the age and the height of the patient)

The age and height of patient 2: Age 51, Height 160cm.

Patient 3 and 5 same patient : Age? seems osteopenic

Patient 3 and 5 are different persons. The age of patient 3 and 5 were 54 and 56, respectively.

Patient 4 , give age? very likely syndromic-delete 

The age of patient 4 was 50. We have deleted the image of patient 4.

Pateint 6 asymmetry ? Lower limb length asymmetry? 

After discussion, we think this may be due to the patient position.